# A Pathological Complete Response to the Combination of Ipilimumab and Nivolumab in a Patient with Metastatic Renal Cell Carcinoma

**DOI:** 10.3390/medicina58030336

**Published:** 2022-02-23

**Authors:** Hana Studentova, Anezka Zemankova, Martina Spisarova, Daniela Skanderova, Zbynek Tudos, Bohuslav Melichar, Vladimir Student

**Affiliations:** 1Department of Oncology, Faculty of Medicine and Dentistry, Palacky University Olomouc and University Hospital Olomouc, I.P. Pavlova 6, 77900 Olomouc, Czech Republic; hana.studentova@fnol.cz (H.S.); anezka.zemankova@fnol.cz (A.Z.); martina.spisarova@fnol.cz (M.S.); bohuslav.melichar@fnol.cz (B.M.); 2Department of Clinical and Molecular Pathology, Faculty of Medicine and Dentistry, Palacky University Olomouc and University Hospital Olomouc, I.P. Pavlova 6, 77900 Olomouc, Czech Republic; daniela.skanderova@fnol.cz; 3Department of Radiology, Faculty of Medicine and Dentistry, Palacky University Olomouc and University Hospital Olomouc, I.P. Pavlova 6, 77900 Olomouc, Czech Republic; zbynek.tudos@fnol.cz; 4Department of Urology, Faculty of Medicine and Dentistry, Palacky University Olomouc and University Hospital Olomouc, I.P. Pavlova 6, 77900 Olomouc, Czech Republic

**Keywords:** pathological complete response, immunotherapy, hypophysitis, renal cell carcinoma

## Abstract

*Background and Objectives*: Complete pathological response after ipilimumab and nivolumab combination therapy in a patient with intermediate prognosis renal cell carcinoma is an uncommon finding. *Case presentation*: A 60-year-old man presented with synchronous solitary metastatic bone lesion and renal cell carcinoma and achieved a complete pathological response after surgical resection of the bone lesion, followed by ipilimumab and nivolumab combination therapy and nephrectomy. The treatment was complicated by hypophysitis and oligoarthritis more than a year after the initiation of the therapy. *Conclusions*: Currently, the combination therapy based on immune checkpoint inhibitors represents the treatment of choice in patients with intermediate- and poor-risk prognosis metastatic renal cell carcinoma. In the present case, preoperative therapy with ipilimumab and nivolumab resulted in a complete pathological response in the renal tumor. Vigilance concerning potential immune-related side effects is warranted throughout the course of therapy and the subsequent follow-up.

## 1. Introduction

Immuno-oncology (IO) is particularly promising in solid tumors with a high burden of immunogenic antigens, such as melanoma and non-small cell lung carcinoma (NSCLC) and presents a great opportunity to control the disease in patients with these tumors. The widespread use of checkpoint inhibitors (ICIs) in medical oncology represents a major shift in the therapeutic approach aiming at cancer cell elimination. The unprecedented clinical success of ICIs has also encouraged further assessment of the efficacy in the neoadjuvant setting. As ICIs have become a widely accepted standard of care in many cancer types [1], and because of growing evidence of pathological complete response (pCR) as a surrogate efficacy endpoint, many clinical trials have started recruitment of patients with various solid tumors employing an anti-PD1 strategy in the preoperative setting. Pathological complete response was reported to strongly correlate with improved overall survival (OS) and/or disease-free survival (DFS) in patients treated with chemotherapy and targeted therapy [2,3,4,5]. Pathological complete response induced by PD-1 blockade might reflect a state of immune activation [6].

The combination of the anti-PD1 antibody nivolumab with the anti-CTLA-4 antibody ipilimumab has shown superior efficacy compared with sunitinib monotherapy in the first-line setting in patients with intermediate and poor prognosis according to the International Metastatic Database Consortium (IMDC) Risk Score for renal cell carcinoma (RCC), introducing a shift in the general therapeutical strategy of the first-line treatment [7]. Moreover, combination immunotherapy with PD-1 and CTLA-4 inhibitors has shown high complete and overall response rates in metastatic setting, and the therapy is now being evaluated in multiple clinical trials to investigate the benefit in the preoperative setting [7,8].

Here, we present a case of metastatic RCC (mRCC) with a solitary metastatic lesion to the bone surgically removed and subsequently treated with combination immunotherapy, resulting in a complete pathological response of the renal primary.

## 2. Case Presentation

A 60-year-old man with an unremarkable medical history presented with pain in the right arm. A computed tomography (CT) scan revealed a metastatic lesion of the humerus and a tumor mass size of 5 cm in the left kidney (Figure 1a). In August 2019, the bone lesion was surgically removed with negative margins, and histology confirmed metastasis of clear cell carcinoma of renal origin (Figure 2a,b). Given the IMDC intermediate-risk classification (<1 year from time of diagnosis to systemic therapy), the combination immunotherapy was initiated in September 2019 with a plan of subsequent surgical therapy of the renal primary. The patient received 4 cycles of nivolumab at 3 mg/kg and ipilimumab at 1 mg/kg given every 3 weeks. Follow-up CT scan showed a change of density in the renal mass (decrease from 76 to 36 Hounsfield units) (Figure 1b) and “ground-glass” changes in basal parts of lungs. The patient complained of xerostomia, but otherwise remained asymptomatic and continued with maintenance nivolumab monotherapy at a flat dose of 480 mg every 4 weeks. In March 2020, substitution therapy with levothyroxine was initiated because of laboratory signs of hypothyroidism. The patient then underwent elective radical nephrectomy in May 2020. Final histological analysis revealed pCR with no residual viable tumor and only dispersed lymphocytic infiltrates along with features of cell death, including cholesterol clefts and interstitial foamy macrophages (Figure 3). The postoperative course was complicated by a prolonged recovery accompanied by asthenia, lethargy, anorexia, and weight loss. Laboratory tests showed hypocorticism and low levels of adrenocorticotropic hormone, suggesting hypophysitis. Since the patient showed no signs of headache or neurological symptoms, only hormone replacement therapy with hydrocortisone was initiated. After recovery from the surgery, the maintenance of nivolumab every 4 weeks was continued. Six months after the surgery, the patient complained about worsening knee pain and had to be examined by an orthopedist. A small amount of serous fluid was removed from both knees, followed by intra-articular methylprednisolone application. Within a month, the stiffening of the knees became disabling and refractory to nonsteroidal anti-inflammatory drugs. Prompt relief after systemic corticosteroid (at a starting dose of 1 mg/kg methylprednisolone) administration with slow titration within 2 months suggested immune-related arthritis. The treatment with nivolumab was permanently discontinued, and the patient remains disease-free 30 months after initial diagnosis.

## 3. Discussion

A case of a patient who initially presented with renal cell carcinoma and a synchronic metastatic lesion in his right humerus is described here. The patient had the solitary bone metastasis surgically removed and started ipilimumab and nivolumab combination therapy. A complete pathological response of the primary kidney tumor was obtained. Immunotherapy-induced hypophysitis was managed by hormone substitution therapy. Subsequently, the treatment was terminated due to the development of arthritis. Although few cases of pathological complete response have been described, to the best of our knowledge, no case of pathological complete response has been reported in a patient with bone metastasis so far, highlighting the potential of the preoperative (neoadjuvant) approach in the era of immunotherapy.

There is currently no FDA-approved systemic therapy in patients with localized or locally advanced RCC. Phase 2 studies and case series addressing this treatment approach with the aim of downstaging the primary tumor using multiple tyrosine kinase inhibitors (MTKIs) have been reported, but results have shown low complete response rates [9]. The preoperative strategy aiming to widen the surgical options resulting in a curative approach was not successful with MTKIs except of occasional case reports [10,11,12]. Recently, Gorin et al. published data on 17 high-risk nonmetastatic RCC patients treated with preoperative nivolumab (only 3 doses of nivolumab, 240 mg, administered intravenously) with 1 patient experiencing pathological response with immune-related features in the removed kidney [13]. In a subset of patients with RCC, ICIs may obviously promote antitumor response by PD-1/PD-L1 inhibition, meanwhile potentially inducing a long-term effect by the elimination of metastatic clones [13]. Several clinical trials, including single-agent (e.g., PROSPER RCC) and dual-agent immunotherapy currently underway, may elucidate the optimal therapeutic strategy in this patient population [14,15].

Regarding the current first-line treatment algorithm, in addition to the combination of ipilimumab plus nivolumab, several other phase 3 trials comparing the combinations of anti-VEGF drugs with ICIs, including axitinib and anti-PD1 antibody pembrolizumab [16], axitinib and anti-PD-L1 antibody avelumab [17], bevacizumab and anti-PD-L1 antibody atezolizumab [18], cabozantinib and anti-PD1 antibody nivolumab [19], lenvatinib and pembrolizumab [20], have reported improved outcomes for the combination therapy over sunitinib monotherapy. An OS benefit has been reported so far for the nivolumab plus ipilimumab and pembrolizumab plus axitinib combinations [7,16]. Although there is currently no prospective trial directly comparing the combination of ipilimumab and nivolumab and the combinations of anti-PD-1/PD-L1 antibodies and VEGF inhibitors, available retrospective data indicate similar efficacy [21]. The combination of ipilimumab plus nivolumab shows one of the highest rates of radiographic complete responses (10.1–12.8%) regardless of the risk group [7]. Optimal patient management and careful patient selection for first-line therapy are of paramount importance affecting the second-line therapy, with a significant impact on survival [22]. The nivolumab and ipilimumab combination immunotherapy, on the other hand, has led to high complete and overall response rates in the metastatic setting [7], and the results of clinical trials in the preoperative setting, such as the NORDIC-SUN trial, are eagerly awaited [8]. In the present case, only a small decrease in the total size of the primary tumor was observed, but the soft-tissue component initially noted was replaced by hemorrhagic fluid and a thin fibrous wall on subsequent CT examination. Most importantly, no viable tumor cells were identified on histological examination. This finding is in accordance with previous reports of radiographic and pathological discrepancy in tumor size. This issue was previously discussed in TKI-treated patients [23] but not as evident as in patients treated with immunotherapy.

Some distinctive patterns of radiographic response have been identified in patients treated with immunotherapy, including pseudoprogression, leading to the development of immune-modified response evaluation criteria in solid tumors (imRECIST), which refined guidelines to assess the clinical benefit of cancer immunotherapy [24]. Recently published data on pathological findings in patients treated with PD-1 blockade describe specific pathological features that distinguish pathological responders from nonresponders, including high numbers of tumor-infiltrating lymphocytes (TILs), neovascularization, fibrosis, cholesterol clefts, and tertiary lymphoid structures [6,25,26]. These changes associated with response appear to reflect immune activation in comparison with coagulation necrosis or hyalinized fibrosis described in tumors treated with neoadjuvant chemotherapy [6]. The role of the pathologist is crucial in the process, regarding adequate sampling and reproducible response scoring in particular.

What could be of concern is the discrepancy between the radiographic and pathological size of the tumor. In one series, only 10% of patients achieved partial response on imaging, and no patient had a complete response in contrast to 45% major pathological responses (mPRs) obtained in the same subset of patients, including three pCRs [6]. This can be explained by an emerging new phenomenon called tumor regression bed that denotes the tumor cell mass replacement by immune cells. Finally, viable tumor cells represent only a small proportion of the entire tumor mass. Distinct responses to immunotherapy have been described not only across a spectrum of different patients but also within individual cases. This may be explained by a heterogeneity of the tumor and the tumor immune microenvironment that represent a dynamic and complex entity with unpredictable behavior [27].

PD-L1 expression is not usually uniform, and heterogeneity of the tumor may be reflected in diverse tumor responses of immunotherapy [28,29]. Factors determining the efficacy of immunotherapy are more complex, and the role of many other factors involved in antitumor immune response has been proposed (e.g., neutrophilic infiltration associated with the upregulation of the VEGF pathway and poor prognosis in RCC). Moreover, increased interstitial pressure, hypoxia, and acidosis induced by neovascularization may represent local factors causing a suboptimal immune response in large primary tumors [30]. In general, the predictive role of PD-1/PD-L1 tumor expression in RCC has not been established in contrast with other solid tumors, including breast, lung, urothelial, where the correlation of PD-L1 expression with response to immunotherapy has been reported [31,32,33,34]. In RCC, no correlation of high PD-L1 expression with response to immunotherapy has been reported so far [35]. The reason why poor-risk RCC patients respond better to ICIs compared with good-risk patients could be associated with higher tumor mutational burden (TMB) [36,37,38]. So far, no reliable biomarkers are available to identify patients likely to have a dramatic response to immunotherapy or, on the contrary, resistance to treatment.

Reports on pathological complete response to ICIs reported to date are scarce. Two cases were reported, including a complete pathological response in an mRCC patient treated with nivolumab after the failure of TKIs [39,40]. Pandey et al. reported a complete pathological response in the kidney and a radiographic complete response in other sites in a poor prognostic mRCC patient treated with the combination of nivolumab and ipilimumab [41]. Similar cases of mRCC patients with a large inferior vena cava tumor thrombus achieving a complete pathological response within the tumor and the thrombus were described [42,43]. Two other cases of exceptional response to the combination of nivolumab and ipilimumab in a left renal mass and significant retroperitoneal and iliac lymphadenopathy in one case and a renal mass with renal vein involvement and extension to the liver in the other case were published lately [44,45]. On the other hand, in a series reported by Singla et al., only 1 patient out of 11 reached a complete pathological response after receiving nivolumab and ipilimumab and subsequent nephrectomy [46]. A longer follow-up is required to demonstrate that a long-term response or even cure was achieved in these patients as anticipated [47,48].

Last but not least, the role of cytoreductive nephrectomy (CN) in the IO era represents a clinical challenge [49,50]. Taking into account the results from the combination trials, the PFS benefit was observed in patients undergoing prior nephrectomy in the ipilimumab plus nivolumab combination and nivolumab plus cabozantinib, with no benefit noticed in pembrolizumab plus axitinib [7,16,19,51]. With regard to OS, a survival benefit associated with prior nephrectomy status was confirmed only in patients treated with nivolumab plus cabozantinib, not in nivolumab-plus-ipilimumab- or pembrolizumab-plus-axitinib-treated patients [7,16,19,51]. However, the analyses did not distinguish between patients with prior nephrectomy in the past versus CN. Singla et al. reported on survival benefit in patients treated with CN plus ICIs versus ICIs alone in a retrospective study from a registry-based cohort of patients [52]. Moreover, a preoperative setting of ICIs was associated with a better outcome in terms of tumor stage and grade compared with patients receiving CN upfront, including 10% patients achieving pCR in the primary tumor [52]. It should be noted that patients undergoing CN upfront had more favorable tumor characteristics. Interestingly, Pieretti et al. reported a better survival outcome in mRCC patients with an intermediate-risk score and achieving metastatic tumor shrinkage of at least 10% after preoperative therapy (TKI, ICI, or both) followed by CN [53]. Nevertheless, there is no clear evidence of CN indication and timing, and the role of CN remains a matter of debate. An individual approach including optimal timing should be discussed in a multidisciplinary team. Defining the role of CN in the era of ICIs warrants prospective validation in clinical trials.

No issue related to the surgical procedure in terms of wound healing was noted in the present case. In the era of TKI therapy, wound healing complications were a concern, and local wound healing complications were described [54]. Immunotherapy may not influence wound healing, but the surgical procedure itself could be a challenge due to fibrotic changes induced by tumor response [28,47,55].

## 4. Conclusions

The present case of RCC with a solitary metastatic lesion to the bone primarily surgically removed and with pathological complete response to preoperative ipilimumab and nivolumab of the primary tumor prompts further investigation of the role of neoadjuvant ipilimumab and nivolumab in patients with advanced RCC.

## Figures and Tables

**Figure 1 medicina-58-00336-f001:**
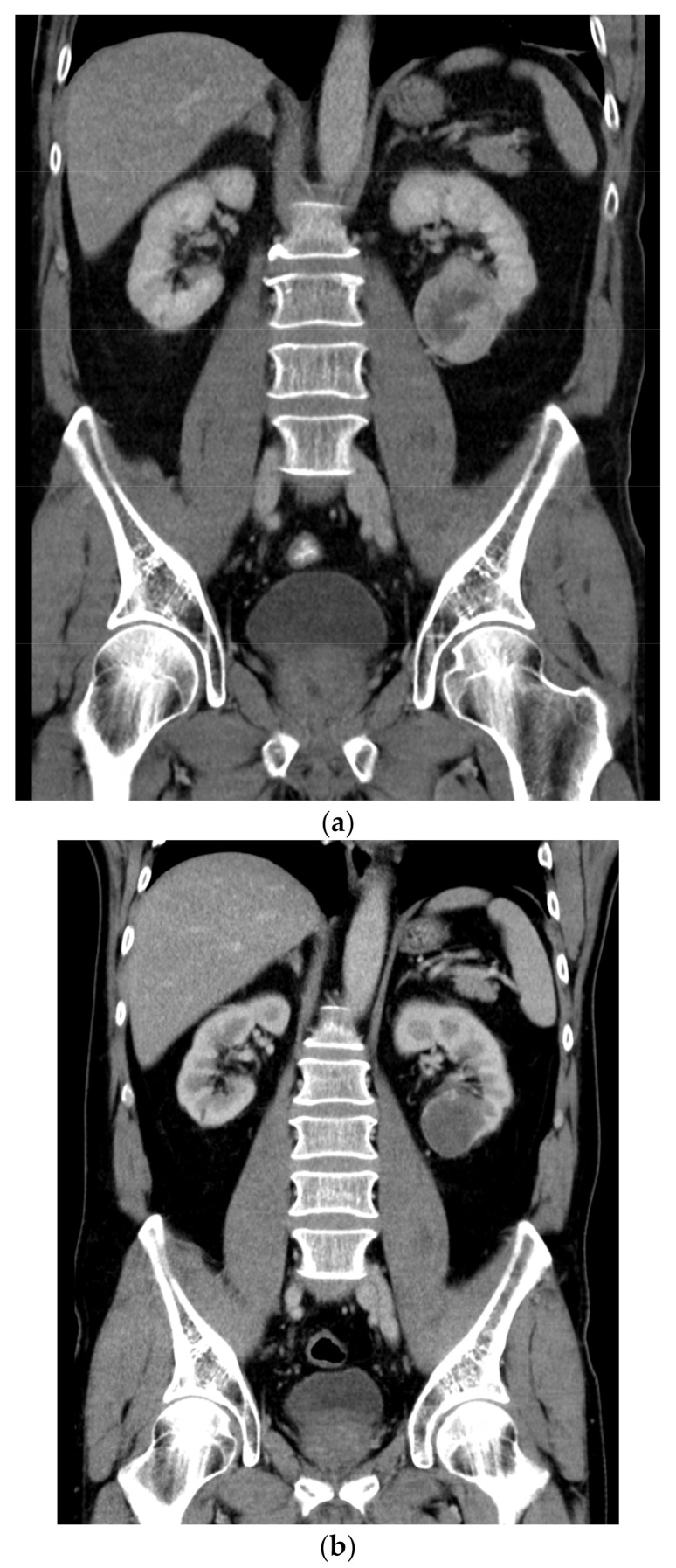
(**a**) Initial contrast-enhanced CT scan in a coronal plane. A tumor on the lower pole of the left kidney contains a heterogeneous enhancing soft-tissue component and a central necrosis. (**b**) Follow-up contrast-enhanced CT scan in a coronal plane. There is only slight regression of the tumor size, but the soft-tissue component is replaced by hemorrhagic fluid and a thin fibrous wall.

**Figure 2 medicina-58-00336-f002:**
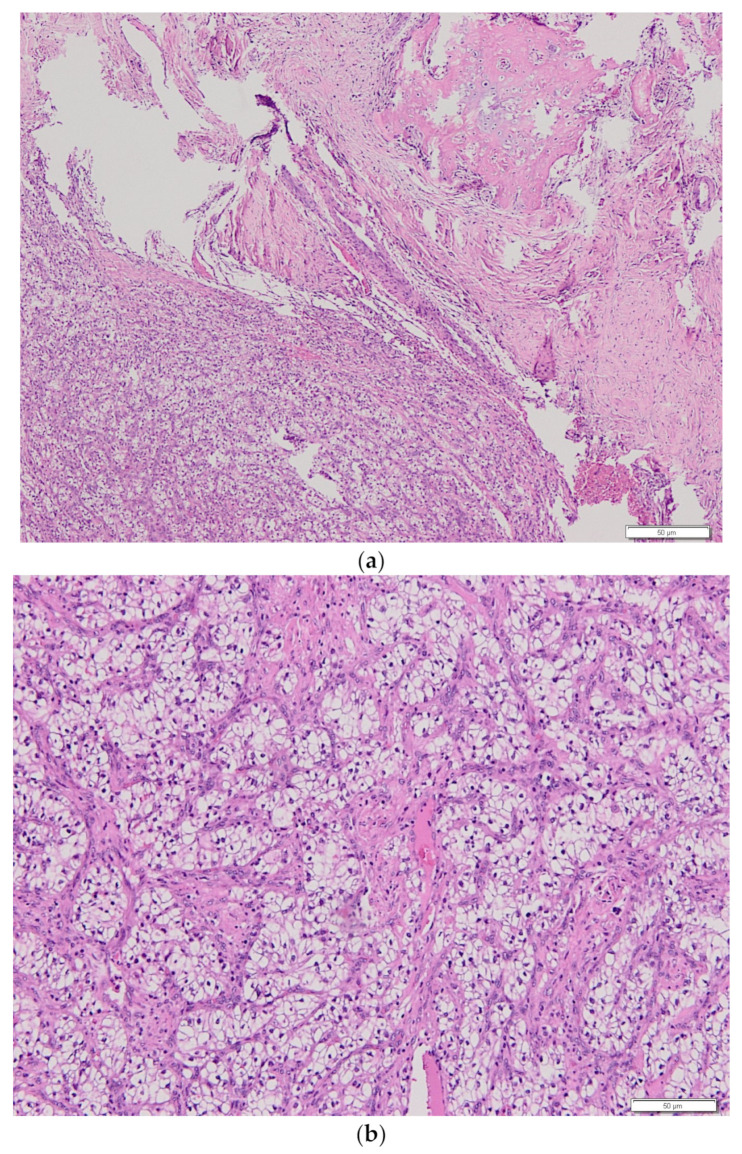
(**a**) Histopathological examination. A metastatic lesion of clear cell carcinoma to the humerus. (**b**) Histopathological examination. A metastatic lesion of clear cell carcinoma to the humerus in detail.

**Figure 3 medicina-58-00336-f003:**
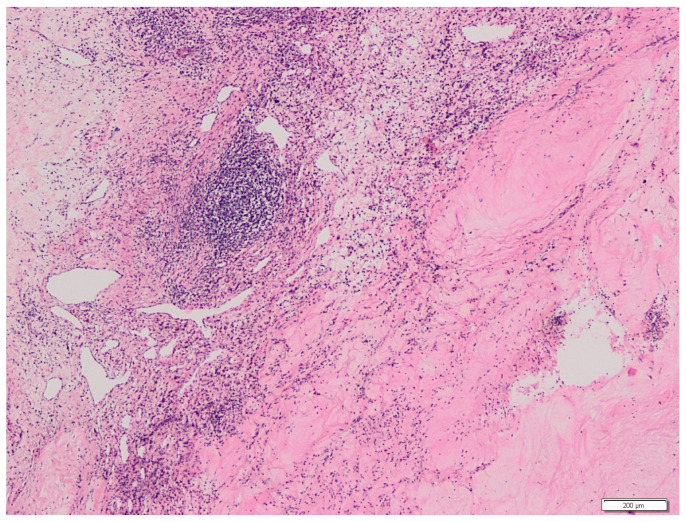
Histopathological examination of the primary tumor after radical nephrectomy showing pathological complete response (pCR). Necrosis, lymphocytic infiltrates, but no residual viable tumor cells are present.

## Data Availability

The data presented in this study are available on request from the corresponding author. The data are not publicly available due to privacy.

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
