# Peer review of "A Pathological Complete Response to the Combination of Ipilimumab and Nivolumab in a Patient with Metastatic Renal Cell Carcinoma"

_medicina, 2022, doi:10.3390/medicina58030336_

Round 1
Reviewer 1 Report
Authors should be congratulated for the intriguing presentation. The widespread use of ICI represents a starting point with interesting future perspectives. Despite several adverse reactions due to ICI, the life span, and the survival earned assess the role of these drugs in patients’ treatment. The manuscript is well written and easily readable, the methodology was robust. Tables and Figures are clear and well-presented. The article adds to the current literature a stimulating proof, but several points warrant a mention:
- Authors should discuss this interesting systematic review on metastatic RCC patients that deserve immunotherapy (PMID: 33291600 DOI: 10.3390/cancers12123634)
- Authors should accurately read this intriguing paper on the role of nephrectomy in metastatic RCC in the immunotherapy era (PMID: 33835666)
Author Response
1. Authors should discuss this interesting systematic review on metastatic RCC patients that deserve immunotherapy (PMID: 33291600 DOI: 10.3390/cancers12123634)
Response: Thank you for your comment. We have mentioned the recommended paper in our text.
2. Authors should accurately read this intriguing paper on the role of nephrectomy in metastatic RCC in the immunotherapy era (PMID: 33835666)
Response: Thank you for your recommendation. We have also included the paper in our manuscript.

Reviewer 2 Report
This is a case study regarding preoperative therapy with ipilimumab and nivolumab resulted in a complete pathological response in the renal tumor.
It does not, in my opinion, fill a specific void in the field.
As this is a case study, the authors should have focused more on topics or the concern that have not been addressed in prior research rather than methodology.
Rather than proposing, it would have been wiser to investigate the role of neoadjuvant ipilimumab and nivolumab in patients with advanced RCC. This could add to the literature.
Some of the citations are outdated. References should be cited consistently throughout the article.
Author Response
This is a case study regarding preoperative therapy with ipilimumab and nivolumab resulted in a complete pathological response in the renal tumor.
It does not, in my opinion, fill a specific void in the field.
Response: We still believe that the case is extraordinary in a way. Achieving complete pathological remission of the tumor is not what we see every day. It is fascinating to observe how effective immunotherapy can be in a subset of patients. Having such a patient standing in front of you is a clinical challenge and we should always seek the optimal treatment strategy considering all points of view.
As this is a case study, the authors should have focused more on topics or the concern that have not been addressed in prior research rather than methodology.
Response: Thank you for your comment. We have revised the discussion part of the manuscript and tried to address some topics in a more detail.
Rather than proposing, it would have been wiser to investigate the role of neoadjuvant ipilimumab and nivolumab in patients with advanced RCC. This could add to the literature.
Response: Thank you for your comment. We have broaden the discussion on neoadjuvant ipilimumab plus nivolumab including adding references.
Some of the citations are outdated. References should be cited consistently throughout the article.
Response: We have revised the references and brought them up to date.

Reviewer 3 Report
Dear Colleagues,
The Manuscript “A pathological complete response to the combination of ipilimumab and nivolumab in a patient with metastatic renal cell carcinoma” by Hana Studentova et al. is a case report revolving around the administration of Checkpoint-inhibitors (Nivolumab 3mg/kg + Ipilimumab 1mg/kg every 3 weeks) as treatment of choice, histologically demonstrating a complete tumor response.
Improvements should be sought in the following items:
- Risk assessment: it would be better to list the item by which the patient was defined as “intermediate risk”
- Content: the setting in which the treatment was administered is the metastatic, intermediate risk, renal cell carcinoma. As of today, nivolumab plus ipilimumab is indicated in various guideline as a treatment option; thus this case report demonstrates that the first-line treatment has been effective, consistent to what originally proposed by Motzer et al. in the phase III trial (NCT02231749) in which a complete response rate (even if described as exploratory) of 9% was achieved. re-elaborate this part of the manuscript would improve the overall intelligibility
- Case structure: defining the precise date in which the patient started the treatment would be a great implementation to treatment timing presentation.
Author Response
Improvements should be sought in the following items:
- Risk assessment: it would be better to list the item by which the patient was defined as “intermediate risk”
Response: Thank you for your comment. The text has been changed accordingly.
Given the IMDC intermediate-risk classification (<1 year from time of diagnosis to systemic therapy), the combination immunotherapy was initiated in September 2019 with a plan of subsequent surgical therapy of the renal primary.
- Content: the setting in which the treatment was administered is the metastatic, intermediate risk, renal cell carcinoma. As of today, nivolumab plus ipilimumab is indicated in various guideline as a treatment option; thus this case report demonstrates that the first-line treatment has been effective, consistent to what originally proposed by Motzer et al. in the phase III trial (NCT02231749) in which a complete response rate (even if described as exploratory) of 9% was achieved. re-elaborate this part of the manuscript would improve the overall intelligibility
Response: The text has been revised and updated as advised.
- Case structure: defining the precise date in which the patient started the treatment would be a great implementation to treatment timing presentation.
Response: The text has been changed accordingly.
Given the IMDC intermediate-risk classification (<1 year from time of diagnosis to systemic therapy), the combination immunotherapy was initiated in September 2019 with a plan of subsequent surgical therapy of the renal primary.
The treatment with nivolumab was permanently discontinued and the patient remains disease-free 30 months after initial diagnosis.

Round 2
Reviewer 1 Report
Authors should be congratulated for the intriguing presentation. The widespread use of ICI represents a starting point with interesting future perspectives. Despite several adverse reactions due to ICI, the life span, and the survival earned assess the role of these drugs in patients’ treatment. The manuscript is well written and easily readable, the methodology was robust. Tables and Figures are clear and well-presented. Authors improved the quality of the manuscript. In conclusion, the manuscript is suitable for a publication.Reviewer 3 Report
Dear Colleagues,
The Manuscript “A pathological complete response to the combination of ipilimumab and nivolumab in a patient with metastatic renal cell carcinoma” by Hana Studentova et al. is a case report revolving around the administration of Checkpoint-inhibitors (Nivolumab 3mg/kg + Ipilimumab 1mg/kg every 3 weeks) as treatment of choice, histologically demonstrating a complete tumor response.
The improvements implemented in this revised version are quite outstanding; it is a pleasure to read a deeper "Discussion" Paragraph, in which Authors sift newly-introduced treatment algorithms. All the flaws previously addressed were overcome; the limitations of the Article design itself are unfortunately retained.
Kind Regards